# Highly Accurate Radar Cross-Section and Transfer Function Measurement of a Digital Calibration Transponder without Known Reference—Part I: Measurement and Results

Jens Reimann *, Anna Maria Büchner, Sebastian Raab, Klaus Weidenhaupt, Matthias Jirousek and Marco Schwerdt

Deutsches Zentrum für Luft- und Raumfahrt e.V., Institut für Hochfrequenztechnik und Radarsysteme, 82234 Wessling, Germany

* Correspondence: jens.reimann@dlr.de

**Abstract:** Active Radar Calibrators (ARC), also called calibration transponders, are often used as reference targets for absolute radiometric calibration of radar systems due to their large achievable Radar Cross-Section (RCS). However, before using a transponder as a reference target, the hardware itself has to be calibrated. A novel method, called the three-transponder method, was proposed some years ago and allows for RCS calibration of digital transponders without using any RCS targets as reference. In this paper, this technique is further refined and applied to a setup utilizing only one digital transponder. The accurate measurement design is described and a novel, elaborated data processing scheme is developed to minimize remaining noise and clutter effects in the data. A comprehensive error analysis is presented in the second part of this paper.

**Keywords:** RCS measurement; ARC; reference target; SAR; transponder; absolute radiometric calibration

## 1. Introduction

Transponders and (polarimetric) active radar calibrators (ARCs/PARCs) are widely used for radar calibration (e.g., [1–3]) as they offer high RCS in a compact design compared to passive targets. Digital transponders are a class of devices in which the received signal is first digitized using an analogue-to-digital converter (ADC) and the analogue transmit signal is regenerated from this digital representation using a digital-to-analogue converter (DAC). This allows for the implementation of potential delays in the digital domain, as well as further signal processing such as frequency shifts, multiple target simulation, or the compensation of the transfer function of the analogue components (e.g., [4,5]).

One of the challenges in building a highly accurate transponder is its end-to-end calibration. In contrast to passive targets, the RCS cannot be simulated with well-established and validated software tools such as HFSS [6] or FEKO [7]. The estimation of the transponder RCS from measurements of all its individual components often leads to unacceptably high uncertainties. This makes an end-to-end calibration approach the most feasible option for low-uncertainty RCS estimation of active devices [8,9].

Many end-to-end calibration techniques for transponder calibration use known passive targets as reference. This transfer of a calibration from one device to another always implies an increase in the calibration uncertainty which one wants to avoid. Furthermore, the calibration of the reference target has to be trusted. In the case of passive reference targets, such as trihedral corner reflectors, the reference RCS originates from theoretical expressions (e.g., a close-form geometric optics solution) or simulations. The necessary simplifications required for deducing a theoretical value have to be validated to prove that they accurately represent the real world.

A novel approach was presented in [10], which allows the RCS of digital calibration transponders to be estimated without a known RCS reference. Also known as the "three

transponder method" (referring to the well-known three-antenna method [11] for absolute gain calibration of antennas), it links the RCS measurement directly to distance measurements (Remark: The unit of RCS is m², often quoted in logarithmic scale as dBm². Hence, its base unit is meter, which can be estimated by distance measurements.). All measurement quantities besides the distance are relative ratios which require no absolute reference. For the absolute distance measurement, it was shown that, for carefully designed measurement setups, the RCS uncertainty due to uncertainties of the distance measurement becomes negligible.

The measurement setup used in the previous publication [10] could not achieve superior accuracy due to issues in its measurement setup leading to large uncertainty contributions from multi-path signal components. Nevertheless, it has proven the feasibility of the novel calibration technique and laid the foundation for this follow-on study.

In this paper, the three-transponder approach will be refined and used to calibrate a single digital transponder with very low uncertainty. The original paper was relying on three transponders while this paper applies the calibration technique to a single device. The other two transponders used in the original approach [10] are replaced by a vector network analyzer (VNA, Rohde & Schwarz, ZVA24) and a passive trihedral corner reflector (inner leg length: 0.9 m), whose RCS are yet unknown, but will also be estimated during the calibration process. The measurement setup and data processing are carefully designed to mitigate uncertainties from multi-path effects as much as possible.

In the first part of this paper, the *three transponder method* is briefly described, followed by a description of the measurement principle. The experimental setup and the implementation of the three involved measurements will also be depicted. Afterwards, the data analysis and processing are presented in detail, after which the final RCS and transfer function are estimated from all three measurement setups. Finally, the results are presented and discussed.

In a second paper, we will introduce a comprehensive uncertainty estimation and the validation of the results using independent data.

## 2. Brief Introduction to the Three-Transponder Measurement Principle

The *three-transponder method* aims to estimate the RCS of digital transponders. For a measurement campaign, a permutation of any two devices involved in the campaign has to be measured. While three identical transponders were used in the original publication, the presented results are based on a combination of one digital transponder, a passive corner reflector (only operated as a radar target), and a VNA (only operated as a radar device).

Taking into account the operational constraints of these devices, the three measurement setups of the campaign can be uniquely defined as follows:

- Setup #1: The transponder is acting as the radar measuring the corner reflector.
- Setup #2: The VNA is acting as the radar, which measures the corner reflector.
- Setup #3: The VNA is measuring the transponder, which is operated as a radar target.

It was already shown in the original publication [10] that at least one device capable of acting both as target and as radar is required, which is the transponder in the given case. This digital transponder must support a target/transponder mode (re-transmitting a received signal) as well as a radar mode (generating a signal and receiving its echo). In both modes the same analogue hardware is involved, with the exception that in the transponder mode, the analogue receive and transmit chains are connected by a digital domain unit (see Figure 1) to route the received signal to the transmit chain. This connection in the digital domain is exactly known and does not introduce additional uncertainties.

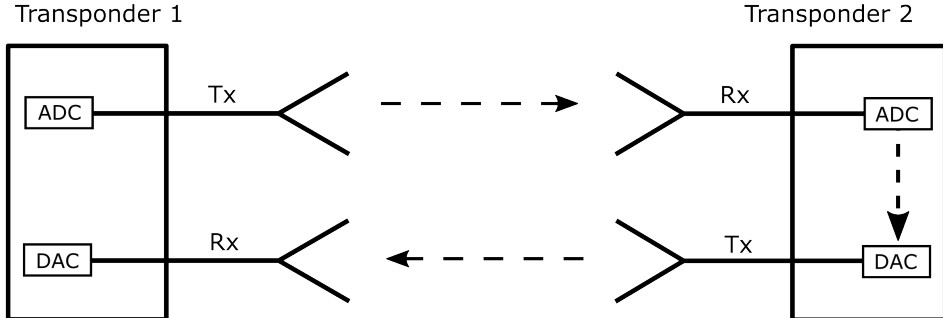

**Figure 1.** Measurement scheme for a transponder in radar (on the left) and in transponder mode (on the right). In transponder mode, the transmitter and the reception chain are connected in the digital domain.

Throughout the manuscript, we refer to the specific equipment which acts in the given setup as the radar (i.e., transponder or VNA) as "radar device". Likewise, the term "target" or "target device" indicates either the transponder or corner reflector in their role as the radar target. When a specific device is met, it is accordingly named "VNA", "transponder", or "corner reflector".

Let us start with the radar equation for a point target, which relates the received power $P_r$ to the transmitted power $P_t$ for a given RCS $\varsigma_A$ of a target A:

$$\frac{P_r}{P_t} = \frac{G_r G_t \lambda^2 \varsigma_A}{(4\pi)^3 R^4} \tag{1}$$

Here, $G_r$ and $G_t$ are the receive and transmit (antenna) gains, $\lambda$ the signals wavelength, and $R$ the distance between radar and target. For a digital transponder B, the RCS can be written as [12]

$$\varsigma_B = \frac{\lambda^2}{4\pi} G_t G_r \tag{2}$$

Combining Equations (1) and (2), the relation between the complex amplitude ratio $a_{AB} = a_r/a_t|_{AB} = (P_r/P_t)^{\frac{1}{2}}\big|_{AB}$ for the device A measuring B, the combined RCS $\varsigma_A \cdot \varsigma_B$, and the equivalent gain $G_A \cdot G_B = (G_{t,A} \cdot G_{r,A}) \cdot (G_{t,B} \cdot G_{r,B})$ can be calculated for each measurement setup as

$$\varsigma_A \cdot \varsigma_B = G_A \cdot G_B \cdot \left(\frac{\lambda^2}{4\pi}\right)^2 = \left(\frac{a_r}{a_t}\bigg|_{AB} \cdot 4\pi R^2\right)^2 \tag{3}$$

From the three individual measurements of a calibration campaign, a system of equations can be formed which is solved for the unknown absolute RCSs $\varsigma_A$, $\varsigma_B$, and $\varsigma_C$.

$$
\begin{aligned}
\varsigma_A &= a_{AB} \cdot a_{AC}/a_{BC} \cdot 4\pi R^2 \\
\varsigma_B &= a_{AB} \cdot a_{BC}/a_{AC} \cdot 4\pi R^2 \\
\varsigma_C &= a_{AC} \cdot a_{BC}/a_{AB} \cdot 4\pi R^2
\end{aligned}
\tag{4}
$$

This scheme is an extension of the linear system of equations given in the original paper [10] and can also be solved for a complex-valued result as it preserves the phase information (with an ambiguity of 180 deg). This solution can also be evaluated over frequency, i.e., solved for several frequency points within the device bandwidth to evaluate the device's transfer function. From this derived transfer function, the RCS (or likewise scattering coefficient) for any (sub-)bandwidth can be calculated.

## 3. Measurement Setup, Constraints, and Geometry

A measurement campaign was conducted to estimate the RCS of a new generation of digital transponders developed by DLR [13] with low uncertainty. This transponder has been developed for future high-resolution synthetic aperture radar (SAR) sensors at X-band with up to 1.2 GHz bandwidth and is the successor of the previous generation C-band "Kalibri" transponders [5].

The transponder features an internal calibration loop to provide a stable instrument [14]: a test signal is generated by the DAC and routed through the transmit chain. Just before the antenna, the signal is guided to the calibration facility and injected into the receive chain where it is recorded by the ADC. This allows most of the RF hardware to be monitored and the hardware drift to be measured. A sophisticated temperature management within the transponder provides stable conditions for the hardware (normally within 0.2 K) and therefore minimizes the hardware drift during operation.

Less sophisticated temperature management is also implemented for the VNA. An additional housing shelters the device from direct sun radiation and rapidly changing thermal conditions. Fans provide stable temperatures within about 10 K and a calibration loop similar to the transponder is also implemented.

As stated in the previous section, three measurement setups for any combination of two devices have to be conducted to perform a full *three transponder measurement* campaign.

A far-field measurement setup in slant range was selected to conduct the calibration campaign. One device was located elevated on a measurement platform on top of an approximately 10 m high building (see Figure 2). The other device was placed on a dolly/railway cart, allowing for a movement along the wave propagation direction to characterize unavoidable multi-path effects ([11] (ch. 12.2.4)), see Figure 3. The railway cart is designed to have very small RCS by having smooth, tilted faces towards the radar source to reflect the incoming energy away from the radar source. Additionally, electromagnetic absorption material was attached to some critical surfaces (see Figure 4). The distortion caused by the remaining backscatter from the railway cart was verified in dedicated measurements of the railway cart without a mounted target.

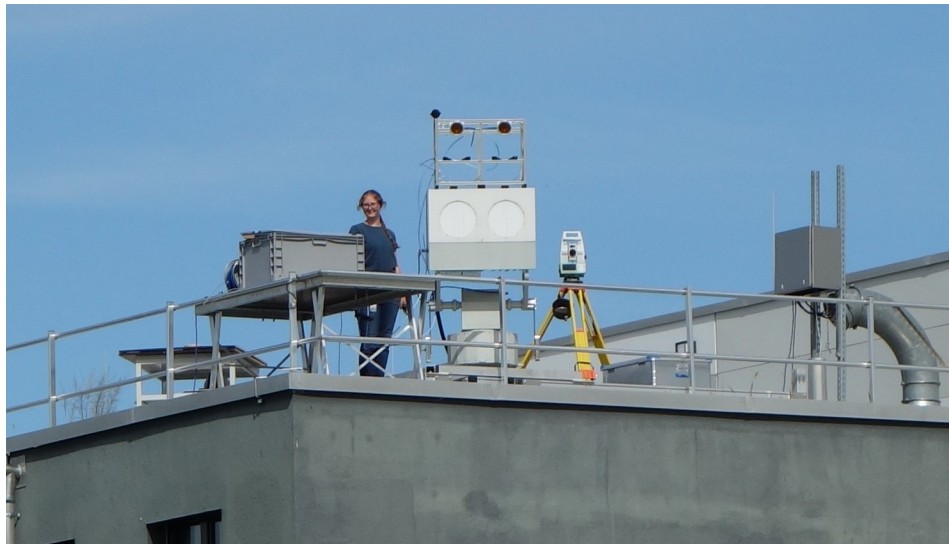

**Figure 2.** Photo of the elevated measurement set up located on the top of a building. The VNA is placed in a temperature-controlled box on the left on a table with its antennas (orange radoms) on top of the transponder (white and gray box on a positioner unit). The geodetic total station is placed on a wooden tripod right of the transponder.

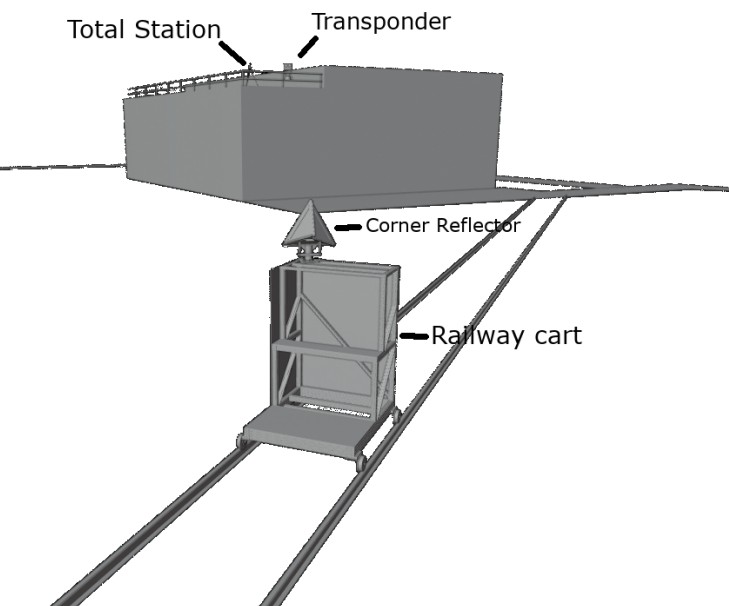

**Figure 3.** Illustration of the measurement geometry. Front: railway cart mounted with a corner reflector; back: building with measurement platform about 70 m away from the target.

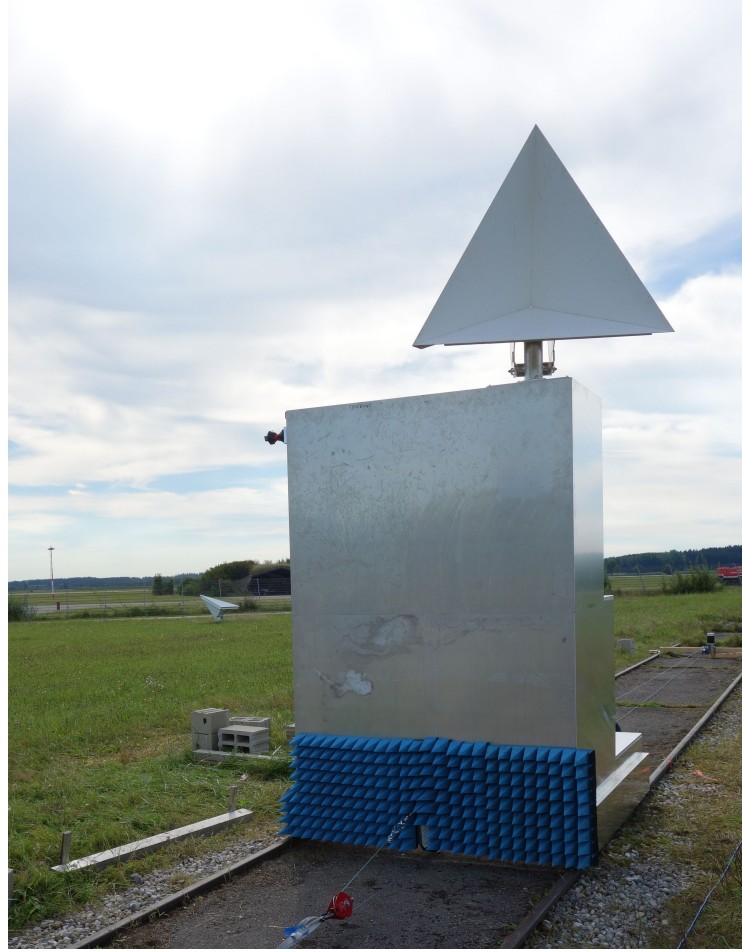

**Figure 4.** Measurement setup of the railway cart including absorbing material to reduce its backscatter.

Each measurement was conducted for various distances between radar and target by moving the railway cart about 10 m around the center measurement distance of approx. 65 m to capture several periods of multi-path signals. The absolute path length difference between the ground-reflected path and the direct path was estimated to be about 1 m for the given geometry (distance and height of the receiver/transmitter, specular reflection on ground). The change in the path lengths (direct vs. specular reflected path) over the 10 m movement distance was more than 33 cm or about $10\lambda$ one way.

The precise distance between the radar device and target was constantly measured using the electronic distance measurement (EDM) functionality of a total station (Leica TS16) deployed in the measurement field. The phase center of the trihedral corner reflector, namely the apex, was measured in reference to a 360° prism (Leica GRZ122) attached to the railway cart, which can automatically be tracked while the cart is moving. The offset between the reference prism and corner reflector apex remains constant, assuming no rotation of the railway cart along the straight track. Hence, the offset is determined only once per setup.

The phase centers of the VNA and the transponder are virtual points within the devices' antennas and can only be measured indirectly. The position of the antennas within their housing is determined from CAD design and/or measurement using a steel ruler. The location of the phase center within the antenna itself was simulated in full-wave using FEKO.

During the measurement campaign, small prisms (Leica GMP111) were attached to all equipment in order to allow the estimation of the real location and rotation of the devices in space and therefore the phase center position of all involved measurement devices. The distance between radar and target for each measurement setup and each railway cart position can then be calculated from the euclidean norm of the 3D coordinates of the phase centers of each of the two involved devices.

The transfer function and RCS of the transponder shall be estimated for the desired transponder bandwidth of 1.2 GHz between 9.2 GHz and 10.4 GHz (ITU frequency allocation for X-band remote sensing). To allow for the capturing of characteristics near the band limits, a measurement bandwidth of 1.5 GHz ranging from 9.05 GHz to 10.55 GHz was selected. This decision was also driven by the fact that the transponder's internal sampling frequency is 4.5 GHz and is hence an integer multiple of the measurement bandwidth. This makes data alignment between frequency domain (VNA) and time domain (transponder) measurements much simpler and does not require any (imperfect) interpolation. Even though the transponder can receive and re-transmit simultaneously (two antennas and dedicated transmit and receive chains), the pulse length of the transponder measurements was set to less than twice the measurement distance (at least 60 m) and was fixed to 350 ns ($< \frac{2 \cdot 60\,\text{m}}{c} \approx 400\,\text{ns}$, where $c$ is the speed of light) to prevent mutual coupling between the transmit and receive path.

The measurement setups for a full calibration campaign are summarized in Table 1.

**Table 1.** Summary of measurement setups used to conduct the calibration campaign.

| Setup | Device on | | Active |
|---|---|---|---|
| | **Roof** | **Cart** | |
| #1 | Transponder | Corner Reflector | Transponder |
| #2 | VNA | Corner Reflector | VNA |
| #3 | Transponder | VNA | VNA |

## 4. Data Analysis

The signal processing and analysis of the measured data is intended to emphasize the wanted signal parts of the measured target while minimizing the effect of noise and clutter. The data processing can be subdivided into three major steps:

1. The data of each individual measurement setup are processed to retrieve the complex-valued spectrum for each device combination (see Figure 5).
2. In a second step, the system of equations given in Equation (4), which combines the results from all measurement setups, is set up and solved for the spectrum of each individual device.
3. In a final processing step, the RCS as seen by a sensor (e.g., a space-borne SAR) can be calculated taking into account the center frequency and bandwidth of the sensor system.



**Figure 5.** Signal processing steps for a single measurement setup.

### 4.1. Processing Strategy

The measurements for the individual targets/radar setups were repeated for different measurement distances (cart positions) to characterize unavoidable multi-path effects. The data are compensated for the measured target position using the total station data to create static conditions for the desired target while making all former stationary echoes (e.g., clutter) move along range (and hence become incoherent). These undesired echoes are therefore alleviated when combining all measurements while the wanted target signature is preserved. This allows for the mitigation of most surrounding echoes and to focus the data on the wanted target. This process will be further detailed in Section 4.3.

The data from the three measurement setups are used as input to solve the set of equations (ref. Equation (4)) for the individual device characteristics. The involved matrix inversion is performed for each frequency point individually to retrieve the device characteristics. At the very end, a time gate can be applied on the data to extract the desired target response and estimate the RCS for a given center frequency and bandwidth.

### 4.2. Data Preparation

Data from two radar devices have to be analyzed, both from the VNA and from the transponder. While the VNA measures the ratio between receive and transmit signals in the frequency domain, i.e., S-parameter, the transponder measures signals in the time domain. The transponder was configured to transmit a linear-frequency-modulated chirp and to receive its respective echo. The ratio between the transmitted signal and received wave can be calculated as

$$S(f) = \frac{Signal_{Rx}(f)}{Signal_{Tx}(f)} \tag{5}$$

in the frequency domain and is similar to the S-parameter recorded by the VNA (Remark: It should be noted that the VNA uses a stepped-frequency approach to estimate the S-parameter, while the transponder uses a chirp. Nevertheless, the retrieved information is considered equal.),

### 4.3. Data Processing for Each Measurement Setup

Once the data from the two radar devices have similar structures, i.e., the time domain data have been compensated for the transmit signal as stated in Equation (5), the frequency-dependent amplitude ratio $a_{XY}$ for a given measurement setup XY can be estimated.

At first, the antenna pattern (or angular RCS variation) for all devices—radar and target—has to be corrected. Due to the measurement setup, a slight change in the incidence angle for the different railway cart positions was inevitable. The variation of the target angle was about 3.3° in azimuth and 1.2° in elevation angle and is known for each instance of time from the position measurements performed by the total station.

Secondly, the direct coupling between the transmit and receive antenna has to be removed. A part of the transmit signal can directly pass to the receive chain at the antenna aperture. This signal is removed by notching the signal over a 5 m window in the time domain. The removal of energy is a valid operation and does not affect the RCS of the to-be-measured target as it is more than 60 m away (even at minimum distance) and no significant energy contribution from the target is expected in the removed data.

Then, the precise target distance can be calculated and the free space loss can be compensated in frequency domain for each frequency point $f$ of the original signal $S_{orig}(f)$ by

$$S_{shifted}(f) = S_{orig}(f) \cdot e^{2\pi j \frac{R}{c} f} \tag{6}$$

where $j$ is the complex unit $j^2 = -1$, $c$ the speed of light, and $R$ the euclidean distance between the radar and the target phase center, see Figure 6b.

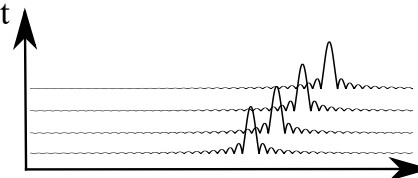

(**a**) Measured data set: the target peak position in fast time ($\tau$) is changing over time (**t**) while the railway cart is moving along the track.

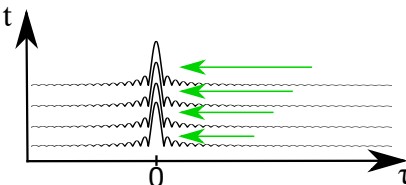

(**b**) Data alignment: each pulse is corrected for the known distance between radar and target, which shifts the target peak to time $\tau = 0$.

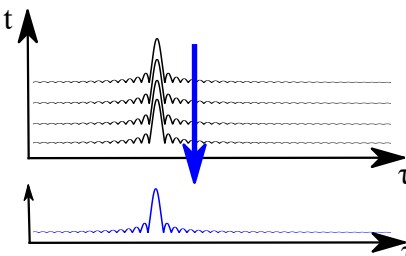

(**c**) Combination: all signals from all railway cart positions are combined into a single measurement signal by means of a Fourier transformation.

**Figure 6.** Simplified signal processing scheme for retrieving the target echo: (**a**) recorded data, (**b**) data alignment, and (**c**) combination step.

The weight and hence the inertia of the railway cart does not allow for a start-stop movement in reasonable time; rather, the cart movement was continuous with about 1.6 cm/s during the whole measurement. This caused a significant movement of the cart during the sweep time of the VNA, i.e., 0.3 s or about 4.8 mm, which is deterministic and also compensated in the data by applying a linear phase ramp in the frequency domain. The movement of the cart during the <400 ns transmission time of the transponder is considered negligibly small and was therefore not corrected.

A small jitter remains between the distance measurement performed by the total station and the radar measurement. It was caused by the not completely deterministic triggering of both measurements. This jitter is estimated in frequency domain by fitting

a linear phase ramp to the data and the variation of the slope is compensated for each railway cart position. The phase has already been corrected to below 360° in the previous step and hence any phase ambiguity is already resolved. Finally the free-space attenuation is corrected for each target position and frequency.

With all deterministic variations removed from the data, the measurements retrieved from different target positions can be combined, see Figure 6c. The data for the various cart positions should now only differ by their multi-path contributions, which can be modeled as a complex-valued sum of the direct signal $s_{direct}(t)$ and various (delayed) multi-path signals $s_{MP,n}(t + \Delta t_n)$ ($n \in [0 \dots N]$) which are attenuated by $c_n$ due to the reduced antenna gain (or RCS) in the direction of the multi-path and the scattering on the ground (see Figure 7).

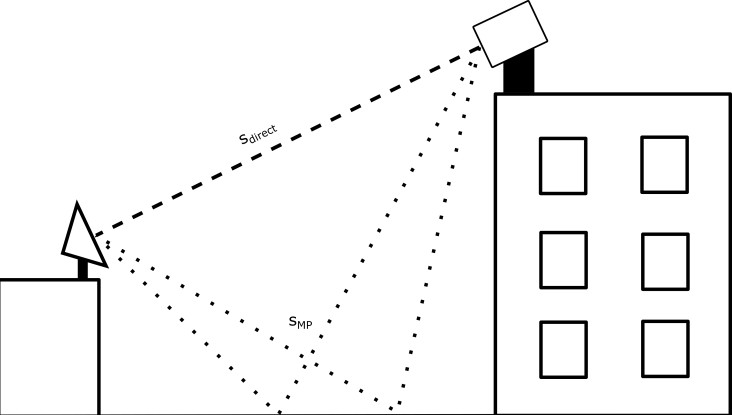

**Figure 7.** Multi-path model used for data analysis.

$$s_{measured}(t) = s_{direct}(t) + \sum_{n}^{N} s_{MP,n}(t + \Delta t_n) \cdot c_n \tag{7}$$

Due to their sinusoidal behavior, the various multi-path components can be separated by a Fourier transformation along the cart positions direction. While the direct path is correctly compensated by the dedicated range measurements, the multi-path components face additional range-dependent phase shifts, which separates them in Fourier space. The direct signal can be extracted at Fourier frequency zero.

In the same step, the incoherent clutter components from the surrounding is spread over the whole frequency space and hence also reduced with the number of measurements (Note: a simple coherent average would have a similar result and would reduced the noise-like clutter by $\frac{1}{N}$ with $N$ the number of measurements along the cart movement).

As an example, the measurement results for the setup *VNA versus transponder* is shown in Figure 8. On the y-axis, the results for the various cart positions are depicted. There are little remaining variations visible in both time domain, Subfigure (a), which would be visible as a range shift of the target peak (yellow color) along the y axis, or in frequency domain, which would manifest itself as a variation of the phase ramp slope for different railway cart positions in Subfigure (c). The final combined signal is presented in sub-plot (d) which is one input used in the next section.

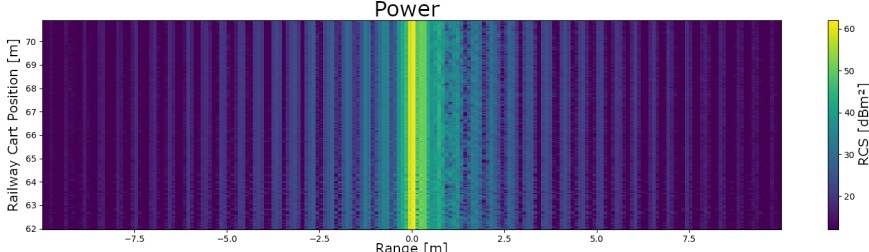

(**a**) Time domain magnitude: x-axis is down range distance and the y axis is the different cart positions. The target is well focused at zero range and no variations along the cart positions are visible.

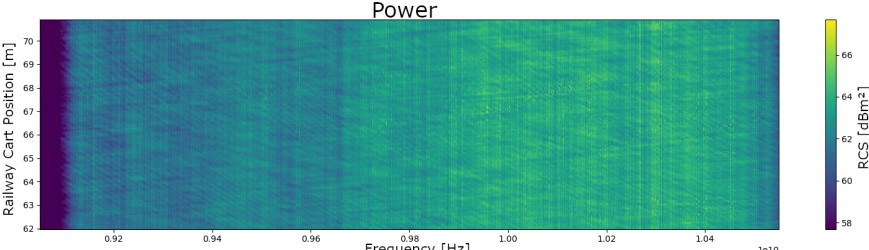

(**b**) Frequency domain magnitude: x-axis is the measured frequency band and the y-axis is the different cart positions. The magnitude changes over frequency (see also the Subfigure (d)), but no significant variations along the y-axis are apparent.

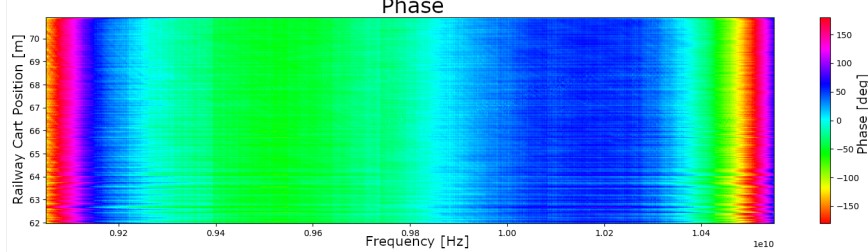

(**c**) Frequency domain phase. Same axes as in Subfigure (b). Again, no significant phase variations are visible for the various cart positions (y-axis).

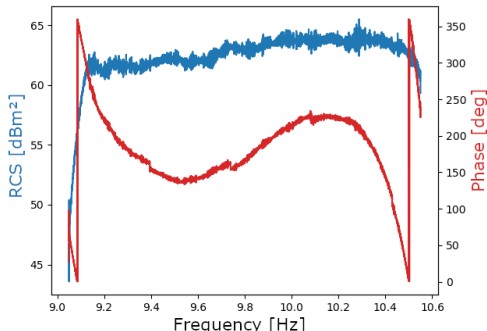

(**d**) Combined complex signal in frequency domain: in blue (left axis): magnitude in dB; in red (right axis): phase in degree.

**Figure 8.** Measurement data from the setup VNA vs. transponder after compensation of known effects, i.e., antenna pattern, free space loss, and target distance, as described in Section 4.3. The measurements for the various railway cart position are shown along the y-axis.

### 4.4. Matrix Inversion and RCS Estimation

After the estimation of the amplitude ratios $a_{AB}$, $a_{AC}$, and $a_{BC}$ for each target/device combination, the system of equations given in Equation (4) can be solved. In contrast to

[10], which solved the system in the logarithmic domain, it is kept in the linear domain here to preserve the phase information.

The system of equations is valid in frequency and time domains for the RCS. The results might be slightly different for frequency and time domains as the measured RCS is superimposed with clutter, noise, and multi-path signals, which may not be completely Gaussian and hence have slightly different characteristics in time and frequency domains (Remark: A Gaussian distributed signal $g(t)$ is invariant to the Fourier transformation: $g(t) = e^{\frac{-t^2}{2}} \stackrel{\mathcal{F}}{\Longleftrightarrow} G(f) = e^{\frac{-f^2}{2}}$.). In the scope of this paper, the frequency domain solution was selected.

The final RCS can now be estimated from the time-domain impulse response function (IRF) using the peak or integral RCS method [15]. The peak RCS method is simpler, but relies on a known shape of the impulse response function. Due to the strong frequency dependency of the transponder transfer function, the peak method should not be used for this device. Still, all results are presented in Table 2.

**Table 2.** Results of the three-transponder measurement campaign. The frequency range 9.2 GHz to 10.4 GHz equals the full SAR bandwidth available in X-Band while the frequency range of 9.5 GHz to 9.8 GHz is the operational bandwidth of the SAR satellites TerraSAR-X and TanDEM-X used during verification.

|  | Transponder [dBm²] | Corner Reflector [dBm²] | VNA [dBm²] |
| --- | --- | --- | --- |
| Peak RCS |  |  |  |
| 9.2 GHz to 10.4 GHz | 62.308 | 34.280 | 47.348 |
| 9.5 GHz to 9.8 GHz | 62.257 | 34.158 | 47.478 |
| Integrated RCS |  |  |  |
| 9.2 GHz to 10.4 GHz | 62.848 | 34.265 | 47.348 |
| 9.5 GHz to 9.8 GHz | 62.268 | 34.137 | 47.464 |

## 5. Conclusions

A comprehensive measurement campaign and analysis of corresponding data was presented to estimate the RCS and the complex frequency response of SAR reference targets. The goal was to derive the RCS of a newly developed transponder with low uncertainty. The selected technique, the adapted *three-transponder method*, allows not only the RCS of the transponder to be estimated, but also the RCS of the involved trihedral corner reflector and the gain of the antennas used for the VNA without the necessity of an external a priori known RCS target.

A comprehensive measurement model was developed to correct the measured data for known effects and to separate the wanted measurement signal from unwanted components such as noise and clutter. A dedicated uncertainty analysis will be presented in another paper.

**Author Contributions:** Conceptualization, K.W., J.R., M.J.; methodology and formal analysis, J.R.; measurement, A.M.B., S.R., J.R.; resources and funding acquisition, M.S.; investigation, J.R., M.J., S.R.; software, A.M.B., J.R.; writing—original draft preparation, J.R.; writing—review and editing, M.S., A.M.B., M.J., K.W., S.R. All authors have read and agreed to the published version of the manuscript.

**Funding:** This research received no external funding.

**Data Availability Statement:** The data presented in this study are available on request from the corresponding author.

**Conflicts of Interest:** The authors declare no conflict of interest.

## Abbreviations

The following abbreviations are used in this manuscript:

| | |
|---|---|
| ADC | Analogue-to-digital converter |
| ARC | Active Radar Calibrator |
| EDM | Electronic Distance Measurement |
| PARC | Polarimetric Active Radar Calibrator |
| RF | Radio Frequency |
| CR | Corner Reflector |
| CTR | Compact Test Range |
| DAC | Digital-to-analogue converter |
| IRF | Impulse Response Function |
| RCS | Radar Cross Section |
| SAR | Synthetic Aperture Radar |
| TR | Transponder |
| VNA | Vector Network Analyzer |

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
