# Peer review of "Highly Accurate Radar Cross-Section and Transfer Function Measurement of a Digital Calibration Transponder without Known Reference—Part I: Measurement and Results"

_remotesensing, doi:10.3390/rs15041153_

Round 1

Reviewer 1 Report

The manuscript follows the efforts at DLR for creating reliable methods for SAR calibration using active calibrators. It is the first of a 2-part paper and likely the most intriguing one will be second, related to the analysis of uncertainties. The proposed technique and the related experiment set up implements an existing technique named "three transponder method" where two transponders are replaced by a vector network analyzer (VNA) and a  trihedral corner. The unknown RCS are estimated by the calibration process.

The manuscript is well written. As far English is concern authors should make consistent the use of "data" as singular or plural. Specific remarks:

Line 44. "The  results ..." I suggest merging the sentence with the one at line 51-52

Line 164: Is the "system of equations" those in eq. (4) ? Please specify in the text

Line 190: What is the meaning of "similar structure"

Fig 6a:  Caption mentions "fast time", the label "range". Please recall the meaning of "fast time" (somewhere) and force consistency between caption and label.

Line 201-203: The meaning of this sentence is not clear to me. Could you please rephrase it ?

Line 230-231: Figure 6 shows that multipath component has been removed. Could authors indicate the magnitude of multipath component that has been eliminated ? I understood that the set up of the railway was necessary to estimate this effect.

Author Response

Thank you for the valuable suggestions. 
The manuscript has been reviewed to use "data" only in its singular form.

Line 44: The sentence is indeed redundant to line 51-52. It has been removed in favor to the later one.

Line 164: A reference to the eq. (4) has been included into the enumeration.

Line 190: The data from the transponder in radar mode are recorded in time domain while the data acquired by the VNA is measured in frequency domain. Section 4.2. describes how to convert them into a common format with "similar structure". 
In order to make this a approach clearer, the sentence  in section 4.3. has been reformulated as follows: 
"[...] have similar structure, i.e. the time domain data has been compensated for the transmit signal as stated in eq. (5), [...]"

Fig 6a: Thank you for pointing out this discrepancy. The caption has been corrected to "down range distance".

Line 201-203: The paragraph has been reformulated: 
"Secondly, the direct coupling between the transmit and receive antenna has to be removed. A part of the transmit signal can directly pass to the receive chain at the antenna aperture. This signal is removed by notching the signal over a 5 m window in time domain. The removal of energy is a valid operation and does not affect the RCS of the to be measured target as it is more than 60 m away (even at minimum distance) and no significant energy contribution from the target is expected in the removed data."

Line 230-231: The multipath uncertainty will be analyzed in the second part of the paper. It is based on the (remaining) variability of the RCS for the different railway cart positions after correcting for all known variations, like free-space loss, antenna pattern, ... . The contribution is changing a bit between the three measurement setups which are needed to conduct the full measurement campaign, but is in the order of 0.04-0.065 dB. 

Reviewer 2 Report

It could be useful if you can insert a picture of the VNA and transponder with their respective antennas. There is an errated end of line at the row 199 and a comma at the  row 204.

Author Response

Thank you very much for your corrections and ideas to improve the paper.

A new photo has been included as figure 4 showing the measurement setup of both, the transponder and the VNA, on the roof of the building with their antennas.

Line 199: The errated end of line will be fixed for the final layout.
Line 204: the isolated comma has been removed

Reviewer 3 Report

This study seems to be a technical report instead of an article. The significances of the content need to be clarified. The disvantagess of the previous approach (the three transponder approach) should be introduced. The gaps to fill with this research still need to be introduced.

Some details in the experiment part are needed to be included.
1. Line 178-179. What is the set of equations for each individual device characteristics? How the matrix inversion is involved?
2. Line 209. How is the data compensated due to the significant movement of the cart for the VNA?
3. Line 214-215. How is the jitter estimated and compensated?
4. Line 223-226. It's suggested that an illustration with the measured data is provided for the separation of multi-path in the frequency domain.
5. Line 231-232. How can we tell from Figure 6 that there are few remaining variations?
6. Line 245-246. How is the transponder transfer function strongly frequency dependent?

Author Response

Dear reviewer, thank you for your suggestions. 
Indeed, the manuscript should be treated as a technical note, because, following the MDPI Remote Sensing regulations, an article must have at least 18 pages.

Thanks to the suggests of the reviewer, the introduction has been extended to show more clearly the differences to the paper which introduced the novel calibration approach.
("The measurement setup used in the previous publication [10] couldn't achieve the superior accuracy due to issues in its measurement setup leading to large uncertainty contributions from multi-path signal components. Nevertheless, it had proven the feasibility of the novel calibration technique and laid the foundation for this follow-on study.

In this paper, the three transponder approach will be refined and used to calibrate a single digital transponder with very low uncertainty. The original paper was relying on three transponders while this paper applies the calibration technique to a single device. The other two transponders used in the original approach [10] are replaced by a vector network analyzer (VNA, Rohde & Schwarz, ZVA24) and a passive trihedral corner reflector (inner leg length: 0.9 m), whose RCS are yet unknown, but will also be estimated during the calibration process. The measurement setup and data processing are carefully designed to mitigate uncertainties from multi-path effects as much as possible.")

1. Line 178-179: A reference to the system of equations given in eq. (4) was added.
2. Line 209: The VNA sweeps linearly through the measured frequency span during the measurement time. Hence, it can be assumed, that during the frequency sweeps of 0.3s a movement of 4.8mm is performed. This is corrected by a linear phase ramp between the lowest and highest frequency component in frequency domain.
The sentence in line 209 has been extended to clarify this approach ("[...] by applying a linear phase ramp in frequency domain.")
3. Line 214-215: The jitter is a random variation of the measurement start time. This is equivalent to a linear phase ramp in frequency domain, which can be estimated for each railway cart position. The variation from the mean is then compensated.
The sentence has been reformulated: "This jitter is estimated in frequency domain by fitting a linear phase ramp to the data and the variation of the slope is compensated for each railway cart position"
4. Line 223-226: This was actually already considered to be included in the manuscript. But even when using a logarithmic scale, the generated plot does not provide any useful information. The magnitude difference between the wanted "target energy" and the "multi-path" component is so high, that it is very hard to illustrate.
5. Line 231-232: Remaining jitter in the data would cause a shift in time domain (x axis) for the different car positions (y axis) in subfigure (a) or a phase slope in subfigure (c).
To help the reader, the sentence has been modified to include a description of the effects caused by remaining jitter: "There are little remaining variations visible in both, time domain, subfigure (a), which would be visible as a range shift of the target peak (yellow color) along the y axis, or in frequency domain, which would manifest itself as a variation of the phase ramp slope for different railway cart positions in subfigure (c)."
6. Line 245-246: The strong frequency dependency originates from the analogue hardware, especially the impendence matching at the ADC. The utilized combined FPGA-, ADC-, DAC-board is a COTS part and was not designed nor optimized for this project.

Reviewer 4 Report

Figure 5 (b)/(c)

In order to correctly realign the recorded echoes, before to combining them into a single echo, have you considered just the information available from the EDM (let's say passively) or have you also tried to search for the position of the echoes (peak or rising edge) of the collected echoes in the Receiving Window? and then realign the echoes according to this supplementary information?. I was wondering if there could be some small differences caused for example by an unexpected inaccuracy of the EDM.

Author Response

Thank you for your question.

The position was mainly retrieved from the EDM. The EDM (and the total station in general) is used for geodesy and has a very high accuracy in the order of a few millimeter over distances of several 100 m (a detailed analysis will be presented in the uncertainty evaluation paper). There is actually a discrepancy between the radar and the EDM distance caused by the different measurement time of both instruments combined with the movement of the railway cart. This jitter is compensated using the phase information of the data (see line 212 in the original manuscript or 225 in the revised version).

Reviewer 5 Report

The three-transponder-method allows for RCS calibration of digital transponders without using any RCS target as reference. ✺ In this paper, three-transponder-method is further refined and applied to a setup utilizing only one digital transponder, which is used for RCS calibration of digital transponders without using any RCS target as reference. The accurate measurement design is described and a novel, elaborated data processing scheme is developed to minimize remaining noise and clutter effects in the data. This paper is well-written and scientifically sound.   I suggest the authors to adjust the fontsize in Fig 1 (too large) and Figs 6 & 7 (too small).

Author Response

The you for your comments. 

We have adjusted the font sizes and increased the size for figure 1 and decreased it for figure 6 and 7 to improve readability.

Reviewer 6 Report

The work is clear and the goals are clearly stated.

I'm concerned with the bibliography: 13 out of 16 references are more than 5 years old. Also, the total number of references can be considered small (16).

The work is completely outside my research field so my review could not be very rigorous.

Author Response

Dear Reviewer, 
we take your concern regarding the bibliography seriously. We can only make conjectures why there are few recent publications within the last years. It might be related to the corona pandemic and the associated contact restrictions which made extensive measurements campaign like the presented one very hard to conduct.
While there is a strong and growing demand on well calibrated space-borne radar data (esp. SAR), the community performing the actual calibration is small and the number of groups having enough personal and financial resources to develop complex calibration devices is even smaller. This limits the number of suitable publications we can refer to while it allows us to support a large number of SAR missions with our reference targets.

Round 2

Reviewer 3 Report

The authors have made improvements related to the issues I'm concerned about. The originality of the work has been clarified much better. The current manuscript is suggested to be accepted.